# Electrospinning-Generated Nanofiber Scaffolds Suitable for Integration of Primary Human Circulating Endothelial Progenitor Cells

**DOI:** 10.3390/polym14122448

**Published:** 2022-06-16

**Authors:** Miguel A. Jiménez-Beltrán, Alan J. Gómez-Calderón, Rafael E. Quintanar-Zúñiga, Daniel Santillán-Cortez, Mario A. Téllez-González, Juan A. Suárez-Cuenca, Silvia García, Paul Mondragón-Terán

**Affiliations:** 1Laboratorio de Medicina Regenerativa e Ingeniería de Tejidos, División de Investigación Biomédica, Centro Médico Nacional ‘20 de Noviembre’—ISSSTE. San Lorenzo 502, 3er Piso. Col. Del Valle, Del. Benito Juárez, Mexico City 03100, Mexico; majb@ibt.unam.mx (M.A.J.-B.); ajgc.022@gmail.com (A.J.G.-C.); dsantillan@ciencias.unam.mx (D.S.-C.); mario.atg91@gmail.com (M.A.T.-G.); 2Unidad de Biotecnología en Prototipos (UBIPRO), Departamento de Fisiología Vegetal, FES Iztacala UNAM, Tlalnepantla de Baz 54090, Mexico; juanantonios55@gmail.com; 3Investigación Clínica, Centro Médico Nacional ‘20 de Noviembre’—ISSSTE. San Lorenzo 502, 2do Piso. Col. Del Valle, Del. Benito Juárez, Mexico City 03100, Mexico; suarej05@gmail.com (J.A.S.-C.); rolasil@yahoo.com.mx (S.G.); 4Coordinación de Investigación, Centro Médico Nacional ‘20 de Noviembre’—ISSSTE. San Lorenzo 502, 2do Piso. Col. Del Valle, Del. Benito Juárez, Mexico City 03100, Mexico

**Keywords:** tissue engineering, endothelial cells, nanofiber scaffolds

## Abstract

The extracellular matrix is fundamental in order to maintain normal function in many organs such as the blood vessels, heart, liver, or bones. When organs fail or experience injury, tissue engineering and regenerative medicine elicit the production of constructs resembling the native extracellular matrix, supporting organ restoration and function. In this regard, is it possible to optimize structural characteristics of nanofiber scaffolds obtained by the electrospinning technique? This study aimed to produce partially degraded collagen (gelatin) nanofiber scaffolds, using the electrospinning technique, with optimized parameters rendering different morphological characteristics of nanofibers, as well as assessing whether the resulting scaffolds are suitable to integrate primary human endothelial progenitor cells, obtained from peripheral blood with further in vitro cell expansion. After different assay conditions, the best nanofiber morphology was obtained with the following electrospinning parameters: 15 kV, 0.06 mL/h, 1000 rpm and 12 cm needle-to-collector distance, yielding an average nanofiber thickness of 333 ± 130 nm. Nanofiber scaffolds rendered through such electrospinning conditions were suitable for the integration and proliferation of human endothelial progenitor cells.

## 1. Introduction

Function restoration of failing body organs and the proper healing of laceration wounds are promises from regenerative medicine [1]. In this context, vascular tissue engineering in the last 3 decades has been developed in the face of the need to replace vascular obstructions and the production of angiogenesis for vascular regeneration. Therefore, developing a completely biological substitute, possessing appropriate mechanical conditions, thrombosis resistance and non-inducer of exacerbated immune response, is broadly desirable [2]. New approaches in artificial blood vessels could positively impact ischemic injury and cardiovascular therapies [3,4]. In this sense, production of functional three-dimensional constructs that mimic the extracellular matrix [5], as well as the development of scaffolds with antibacterial properties [6], represents a major challenge for tissue engineering and regenerative medicine [7].

Vascular constructs are miscellaneous and their complexity may vary based on each of their components. Some constructs have combined biopolymers and selenium nanoparticles [1,8]. Other scaffolds are based on cellulose acetate and polyvinyl alcohol containing magnetite nanoparticles or graphene oxide [9].

The nature of each component, either natural or artificial, as well as the loading particles associated, are crucial for physical or biomechanical properties of the construct such as scaffold degradation time, malleability (handling) and/or cell integration [10,11,12]. For instance, natural polymers cannot be easily manipulated, whereas synthetic polymers are more malleable under experimental conditions, but they might contain chemical residues [13,14].

The electrospinning technique is a process that uses electrostatic force to obtain nanofiber arrays from natural or synthetic polymers, resembling the extracellular matrix, with the advantage to control several morphologic characteristics during scaffold production, rendering more optimal conditions for cell integration, proliferation, migration and/or differentiation [15,16,17,18,19,20]. However, low cellular adaptation to the biomaterial scaffold constitutes a frequent limitation; hence, it is necessary to optimize biomaterials into constructs eliciting cellular attachment, integration and eventual accomplishment of physiological requirements of the organism [21,22].

Likewise, vascular cell integration to scaffolds is a key point within functional and cytocompatibility evaluation. Historically, Human Umbilical Vein Endothelial Cells (HUVEC’s) have been widely used in several reports; nevertheless, the findings obtained with these cells may be limited only for basic research, since they are not very suitable for vascular transplants in a clinical landscape. On the other hand, human Endothelial Progenitor Cells (hEPCs) are hematopoietic cells, sharing immunophenotype CD34^+^, CD133^+^ and/or CD31^+^ [23,24,25,26]. hEPCs may be extracted from bone marrow compromising the safety of the patient, with the inconvenience of a limited collection which may also be methodologically complex.

A considerable percentage of hEPCs (0.01 to 0.0001%) may be isolated from peripheral blood, representing challenges but also technical advantages, such as the possibility to collect circulating hEPCs using minimally invasive techniques, with subsequent in vitro cell expansion, as well as the low cost related to this cell derivation method. In addition, optimized conditions of electrospinning parameters, rendering a natural polymer (denatured collagen) suitable for integration of primary human EPCs from peripheral blood, would contribute to obtaining a biological vascular construct suitable for autologous transplant.

Several studies have focused on the morphological and physico-chemical properties of nanofibers obtained by the electrospinning method, although they have not concomitantly explored the functional ability to integrate cells or to evaluate the cytocompatibility of their components. In addition, only few studies have characterized scaffolds suitable for human EPCs integration. Considering natural interactions between the scaffold and cells, it is relevant to evaluate not only morphological characteristics of the obtained nanofibers by electrospinning, but the specific parameters rendering scaffolds with the most optimal features eliciting cell interaction and integration. Therefore, in this study we described nanofiber characteristics obtained under different electrospinning parameters of partially degraded collagen (gelatin), and explored the optimal condition for minimal-aberrations nanofibers which are also suitable for the integration of primary human EPCs derived from human peripheral blood.

## 2. Materials and Methods

Cell culture and all the experiments were performed under strict safety measures, working in a biosafety level II culture hood, and following standardized operational procedures, based on international recommendations, in order to avoid cell culture contamination. The present study was approved by the Institutional Research, Ethics and Biosafety Committees of the National Medical Center ‘20 de Noviembre’ ISSSTE (protocol approval number 077.2016). Authorization and signed informed consent were obtained prior to collecting biological samples.

### 2.1. Electrospinning Set Up and Nanofibers Generation

Electrospinning was carried out using an in-house-produced set consisting of: (1) an infusion pump (Cole-Parmer 200), (2) a rotating mandrel and (3) a high voltage power source (Spellman CZE1000R) (Figure 1), adapted to the specific size dimensions to fit in the destined location. Nanofiber substrate of partially degraded collagen (gelatin) was prepared as follows: denatured collagen (Sigma Aldrich, St. Louis, MO, USA) was suspended in glacial acetic acid solution (Sigma Aldrich, St. Louis, MO, USA) at final concentration of 10% (*w*/*v*), and maintained under room temperature at constant stirring for 48 h. Several working parameters for electrospinning were evaluated, ranging from: voltage, 10 to 15 Kv; rotation speed of collector, 1000 to 3000 rpm; flow rate of collagen suspension, 0.06 to 1 mL·h^−1^; distance tip of needle to collector, 10 to 12 cm. Unless otherwise stated, all experiments were performed as triplicate independent experiments.

### 2.2. Scaffold Nanofiber Assessment

Nanofiber scaffolds obtained from electrospinning were fixed in 4% paraformaldehyde diluted in PBS pH 7.4 free of Ca^++^ and Mg^++^ and dehydrated by increasing concentrations of ethanol; then, scaffold preparations were analyzed by scanning electron microscope JEOL, JSM-6380LV. Microphotographs of representative areas were acquired, and further analyzed regarding diameter, shape and orientation of nanofibers, using Digital Imaging with ImageJ software v 1.44 (NIH, Bethesda, MD, USA). A total of 150 fibers were evaluated for each condition.

### 2.3. hEPCs Isolation and Culture

Fifty milliliters of venous peripheral blood were independently collected from 5 healthy volunteers, aged 25–40 years old, in order to derive hEPCs. First, mononuclear cells were separated by density gradient. Briefly, the blood sample was diluted (1:2 *v*/*v*) with PBS pH 7.4 without Ca^++^/Mg^++^ (Gibco), and further added Histopaque-1077 solution (Sigma) (3:7 *v*/*v*). After 30 min of centrifugation, 500 g at 18 °C and 3-step washes, obtained mononuclear cells were resuspended and cultivated in T-25 flasks pretreated with type I collagen (Gibco, 50 mg/mL) with culture medium Endothelial Basal Medium-2 Bullet kit (Lonza EMB-2 MV); supplemented with 20% FBS, 0.2 mL hydrocortisone, 2 mL hFGF-b, 0.5 mL VEGF, 0.5 mL R3-IGF-1,0.5 mL ascorbic acid, 0.5 mL hEGF, 0.5 mL Gentamicin/amphitricin-1000U, and 0.5 mL heparin. After 72 h (day 3) of culture, a partial change of culture medium was made; afterwards, total renewal of culture medium was carried out every 48 h, for the next 10–15 days. Once the first colonies of hEPCs appeared (after 10–15 days of culture approximately), the amount of FBS was decreased to 10% in the culture medium for hEPCs’ further maintenance. When hEPCs’ confluence of 75% was achieved, cells were subcultured with a seeding density of 1.8 × 10^5^ cells·cm^−2^ in T-25 flasks pretreated with type I collagen. Cell cultures were incubated at standard culture conditions (37 °C, 5% CO_2_), with medium replacement every 48 h and passaged when 75% confluence was achieved.

### 2.4. Immunocytochemistry

Cell immunophenotype was determined using the following primary antibodies: anti-CD133 (Miltenyi Biotec, Bergisch Gladbach, Germany) and anti-CD31 (Genetex, Irvine, CA, USA), both at 1:300 *v*/*v* dilution in PBS and 60 min incubation at room temperature. After 3 washes with PBS, secondary antibody anti-TRITC (Abcam, Cambridge, UK) was added, and DAPI incubation was used for nuclei stain. Immunopositive fluorescent cells were identified using an Olympus IX71 inverted epifluorescence microscope.

### 2.5. Scaffold–Cell Integration

Optimized scaffolds with the lower amount of structure aberrations during electrospinning were tested for interaction for hEPCs culture for cell integration and cytocompatibility analyses. The hEPCs were used between passage 3–4 after isolation, expansion and characterization procedures. hEPCs were seeded on the nanofiber scaffold at a density of 1 × 10^4^ cells·mL^−1^. Cell integration was evaluated after 6 days of hEPCs culture on nanofibers through scanning electron microscope (SEM). Then, the microphotographs were colorated with the software photoshop using the function layer.

### 2.6. Proliferation Kinetics Assay

For cell proliferation assay, a number of 1 × 10^4^ hEPCs·cm^−2^ were seeded in culture plates in triplicate for 168 h, without medium replacement. Then, the number of cells·area^−1^ (cm^2^) was manually counted. The cell viability index (%) was evaluated every 24 h during a total of 7 days (168 h) of culture, using trypan blue exclusion test. Quantification of specific growth rate (*μ*), as well as the doubling time (*t*_d_), was calculated according to the following equations:


*Specific Growth Rate (µ):*

$$µ=\frac{\left(In\left(\#Final\;Cells\right)-In\left(\#Original\;Cells\right)\right)}{\left(Final\;Time-Zero\;Time\right)}$$




*Doubling Time (t_d_):*

$$t_{d}=\frac{\left(In2\right)}{µ}$$



### 2.7. Statistical Analysis

One-way ANOVA and Tukey test, as well as unpaired *T*-test, were performed to compare nanofiber characteristics obtained with different electrospinning parameters. *p*-value < 0.05 was considered statistically significant.

## 3. Results

### 3.1. Electrospinning

Electrospinning was carried out using the equipment previously described in the Methods section. The resulting scaffolds were functionally characterized using the hEPCs isolated from peripheral blood.

Regarding the ultrastructural assessment of the scaffolds, nanofibers were morphometrically evaluated from 5000× magnification scanning electron microphotographs, and image analysis (Image J software, NIH) evidenced structural variations in fiber diameters according to each experimental condition, as well as the presence of “necklace beads” formations (Figure 2).

#### 3.1.1. Voltage and Infusion Rate Variation

During voltage variation assays (Table 1, Figure 2A) a significant negative correlation between voltage and the fiber diameter size was observed for 10 and 15 kV, with an average fiber diameter of 743 ± 295 nm and 443 ± 295 nm, respect ively, under 0.8 mL/h infusion flow rate (*p* < 0.05). Of note, no nanofiber formation was observed at 8 KV.

Likewise, the partially degraded collagen infusion rate during electrospinning affected the fiber diameter (Figure 2B). Infusion rate ranging between 0.6 mL/h and 0.06 mL/h at voltage 15 kV resulted in a fiber diameter of 307 ± 130 nm and 267 ± 122 nm, respectively, meanwhile for 10 kV under these infusion rates (0.6 mL/h and 0.06 mL/h), significantly different fiber diameters of 692 ± 152 nm and 703 ± 131 nm, respectively, were produced (*p* < 0.05). Then, 15 kV with an infusion rate of 0.06 mL/h working parameters was used for further experiments as this condition reduced wasting resources and resulted in more homogeneous fibers (267 ± 122 nm).

#### 3.1.2. Voltage and Rotation Variation

Once the voltage was assessed and infusion rate was established, different collector rotation speed conditions (1000 rpm, 2000 rpm and 3000 rpm) were tested (Figure 2C). Higher revolutions resulted in smaller fiber diameters (*p* < 0.05), either at 10 kV or 15 kV voltage conditions. A collector speed rate of 2000 rpm was chosen as the final working parameter for further experiments because the resulting fiber diameter (281 ± 193 nm) was between optimal ranges.

#### 3.1.3. Voltage and Needle Distance Variation

Finally, the distance between the tip of the needle and the collector was assessed (Figure 2D). At a fixed infusion rate of 0.06 mL/h and collector speed of 2000 rpm, different voltages (10 kV and 15 kV) and needle-to-collector distances (10 cm and 12 cm) were evaluated. A significant difference was observed upon voltage variation, but not for the variation in needle-to-collector distance (*p* < 0.05). We decided to fix 10 cm needle-to-collector distance as the working parameter for further experiments.

Regarding structure aberrations, “rosary bead” shapes (black arrows) were observed under scanning electron microscopy (SEM). Furthermore, specific conditions of 10 kV, 0.06 mL/h, 2000 rpm and 10 cm needle-to-collector distance resulted in an average fiber diameter of 864 ± 510 nm and “rosary bead” formations found every 0.011 µm^2^ (Figure 3A,B). The increase to 12 cm in the needle-to-collector distance resulted in a mean fiber thickness of 604 nm and structure aberrations found every 0.071 µm^2^ (Figure 3C,D). Interestingly, the condition of 15 kV, 0.06 mL/h, 1000 rpm and 10 cm needle-to-collector distance yielded a mean fiber thickness of 403 ± 286 nm and structure aberrations found every 0.79 µm^2^ (Figure 3E,F), whereas the increase to 2000 rpm yielded a mean fiber thickness of 280 ± 193 nm and structure significantly reduced the size and the number of aberrations was sparing (Figure 3G,H).

### 3.2. Cell Phenotype Characterization

Once the hEPC culture was established, after subculture a typical shape in the monolayer of the cultured hEPCs was observed. Evidence of an endothelial progenitor cell was confirmed by immunocytochemical expression of CD31^+^ and CD133^+^ proteins. Such markers appeared after day six and eight (144 to 192 h) of culture. hEPCs were characterized by colony expansion and typical “cobble” morphology. Endothelial phenotype was predominant at day 12 (288 h) (Figure 4A) observing 95.15% and 97.69% of CD31- and CD133-positive cells, respectively. The estimated efficiency of the derivation method for hEPCs obtained after 12 to 14 days of cell culture was between 1.8 × 10^6^ and 3.2 × 10^6^ hEPCs per 50 mL of peripheral whole blood.

### 3.3. Proliferation Kinetics

Cell proliferation was characterized by the following distinguishable phases: (1) adaptation, from 0 to 48 h; (2) exponential growth, from 48 to 120 h; and (3) maintenance, from 120 to 144 h. A phase of cell death, consecutive to the maintenance phase, was evidenced by a significant drop in the number of hEPCs, and further verified through viability assay. In general, hEP-cell-specific growth rate was estimated as 0.03 cells∙h^−1^, and a doubling rate (t_d_) of 23.10 h during the exponential growth phase.

### 3.4. Cell Integration to the Collagen–Nanofiber Scaffold

Scaffolds with the lower amount of structure aberrations during electrospinning (15 kV, 10 cm, 2000 rpm, 0.06 mL/h) were tested for interaction with the cell culture. Established hEPCs were cultured on nanofiber scaffolds, showing cellular integration to the scaffold; moreover, integrated cells tended to align towards the nanofiber direction (Figure 5).

## 4. Discussion

Given the intimate and necessary relationship between cells and the extracellular matrix (ECM), the present work explored some morphological characteristics of a biopolymer scaffold that functionally mimics ECM, as well as the ability of hEPCs to integrate into the biopolymer composed of partially degraded collagen nanofibers. Our main finding was that subtle modifications in the electrospinning parameters directly impacted nanofiber diameter and morphology. Consistently, other groups have reported similar observations [26,27,28].

Electrospinning allows controlling the fiber diameter in the scaffold, which is important for the implantation design and tissue target. Fiber diameter goals in our study ranged between 50 and 500 nm, since fibers within this range are able to interact with proteins and polysaccharides in the native ECM of the vascular tissue [29,30]. Similar to other studies using collagen scaffolds, we used acetic acid as a solvent for partially degraded collagen, which resulted in efficient electrospinning and cytocompatibility of nanofibers.

Control of fiber diameter is relevant because it affects mechanical properties of the scaffold, as reported by Bölgen et al., 2005 [31]. Furthermore, this effect is constant and independent from materials used for the scaffold [31,32,33,34]. In this sense, voltage application represents a useful parameter to modulate fiber diameter. Consistent with other studies [34], we observed a negative correlation between voltage and fiber diameter. On the other hand, needle-to-collector distance has been described to exert dominant effects on the alignment and thickness of the fibers in the scaffold [35,36,37]; however, we failed to achieve significant effects of needle-to-collector distances on fiber characteristics. This may be due to variations in the whole tested conditions, suggesting the need to consider further testing conditions to better characterize this parameter. Other studies [38] have described that fiber diameter is negatively related to needle-to-collector distance and directly related to the percentage of the polymer present in the solution.

Besides the benefit of low-cost polymers used to produce the scaffold, nanofiber configuration obtained using this technique promoted specific cell alignment. Early after hEPCs placement over the electrospinning-generated scaffold, the cells tended to align themselves acquiring a similar arrangement as nanofibers, suggesting a potential control of cell direction by the use of this technique. This effect may be explained since the particular nanofiber alignment resembles dynamic characteristics naturally occurring in the extracellular matrix, which orchestrate several biological processes such as EPC growth, replication or differentiation. In addition, this property of EPCs alignment on gelatin nanofibers is particularly relevant for vascular tissue engineering since it may promote migration, cell-to-cell contact and molecular communication, more closely resembling the natural endothelium structure and function.

A comprehensive characterization of biopolymers usually includes structural analyses based on XRD, EDX and FR-IR. Likewise, the components of nanofibers may be further characterized by comparative ultrastructural analyses, FTIR spectra and differential scanning calorimetry thermograms, using appropriate controls for the biomaterial obtained. Although these validations are highly desirable, the present study represents an initial approach where the main target was the optimization of partially degraded collagen nanofiber morphologies, by modifying electrospinning parameters in order to obtain optimized scaffolds able to integrate and maintain hEPCs derived from peripheral blood. Future research will be focused on a deeper characterization of scaffold structural properties, which may favor hEPCs interaction and be more suitable for vascular tissue engineering preclinical assays [39,40,41].

The findings of the present study provide useful information regarding the qualitative characteristics of collagen nanofibers that may be obtained under several combination working parameters. Moreover, the hEPCs experiments performed demonstrate that the morphological optimization of the scaffold is relevant for the functional characteristics regarding cell interaction and regulation of cell alignment; particularly in human-blood-derived EPCs which have become essential within the field of tissue engineering, given the potential implications for vascular biology and vascular regenerative medicine.

## 5. Conclusions

According to our data, the electrospinning parameters of 15 kV, 0.06 mL/h, 2000 rpm and 10 cm needle-to-collector distance yielded optimal nanofiber morphology, which was cytocompatible and able to integrate human-blood-derived endothelial progenitor cells. Our study also suggests that electrospinning elicits the production of low-cost natural polymer scaffolds, with the possibility to modify several morphological characteristics in the nanofibers, which potentially impact in the hEPCs’ overlay.

## Figures and Tables

**Figure 1 polymers-14-02448-f001:**
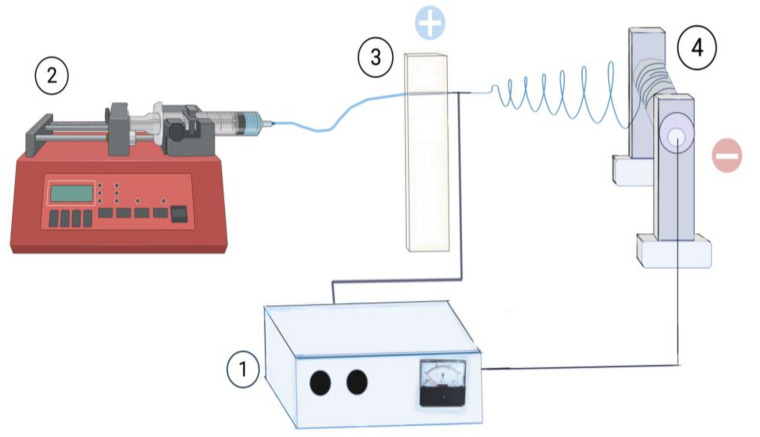
**Description of the electrospinning system.** The parts of the equipment and their location are shown. (1) High-voltage power supply; (2) infusion pump; (3) needle with horizontal movement; (4) manifold with rotating chuck.

**Figure 2 polymers-14-02448-f002:**
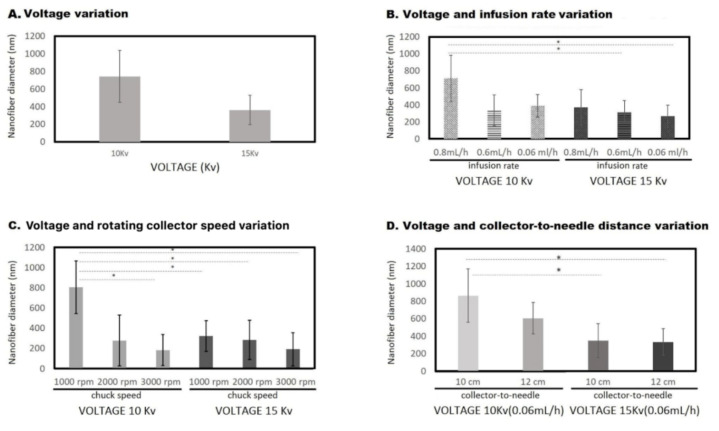
**Nanofibers’ diameter and electrospinning technique.** The figure shows the nanofiber diameters obtained after the variation in the following working parameters: (**A**) Voltage; (**B**) Voltage and infusion rate; (**C**) Voltage and rotating collector speed; and (**D**) Voltage and collector-to-needle distance. (*) Statistical difference *p* < 0.0001.

**Figure 3 polymers-14-02448-f003:**
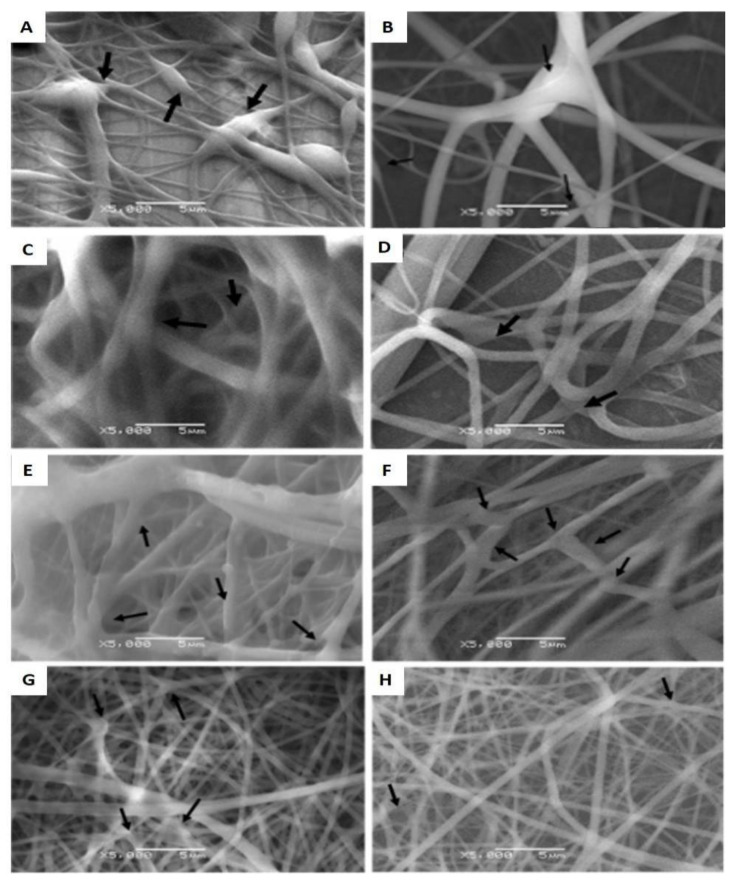
**SEM images of nanofibers morphology**. (**A**,**B**) Particle/aberration within the fibers, observed at condition of 10 kV, 0.06 mL/h, 2000 rpm and 10 cm needle-to-collector distance. (**C**,**D**) The panels show partial particle formation at a condition of 10 kV, 0.06 mL/h, 2000 rpm and 12 cm needle-to-collector distance. (**E**,**F**) Fibers obtained at spinning conditions of 15 kV, 0.06 mL/h, 1000 rpm and 10 cm needle-to-collector distance. (**G**,**H**) Fibers obtained at spinning conditions of 15 kV, 0.06 mL/h, 2000 rpm and 10 cm needle-to-collector distance. Black arrows indicate the presence of aberrations. The scale bar represents 5 microns.

**Figure 4 polymers-14-02448-f004:**
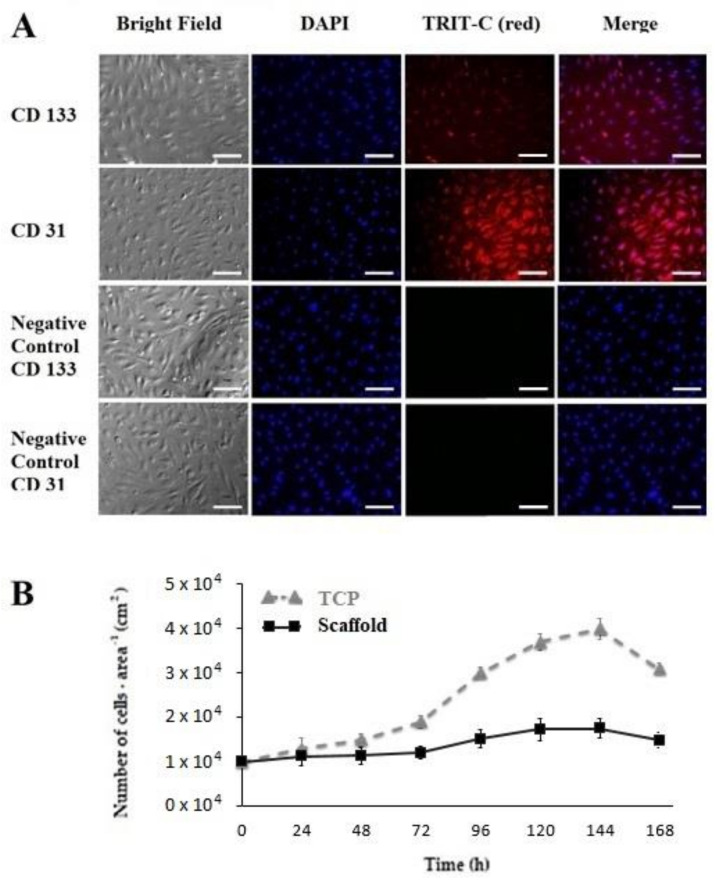
**Immunophenotype and proliferation kinetics of EPCs.** (**A**) Immunophenotype of EPCs obtained after 9 days of subculture. Morphology at bright field, DAPI nucleus stain, CD133 (green), CD31 (red) and merge are shown. (**B**) Proliferation kinetics hEPCs during 7 days of culture, either on TCP (control, growth 0.03 cells∙h^-1^ and a doubling time of 23.10 h during the exponential growth phase) or scaffolds (growth 0.0056 cells∙h^-1^ and a doubling time of 123 h during the exponential growth phase). Cells showed specificity; the scale bar represents 100 microns.

**Figure 5 polymers-14-02448-f005:**
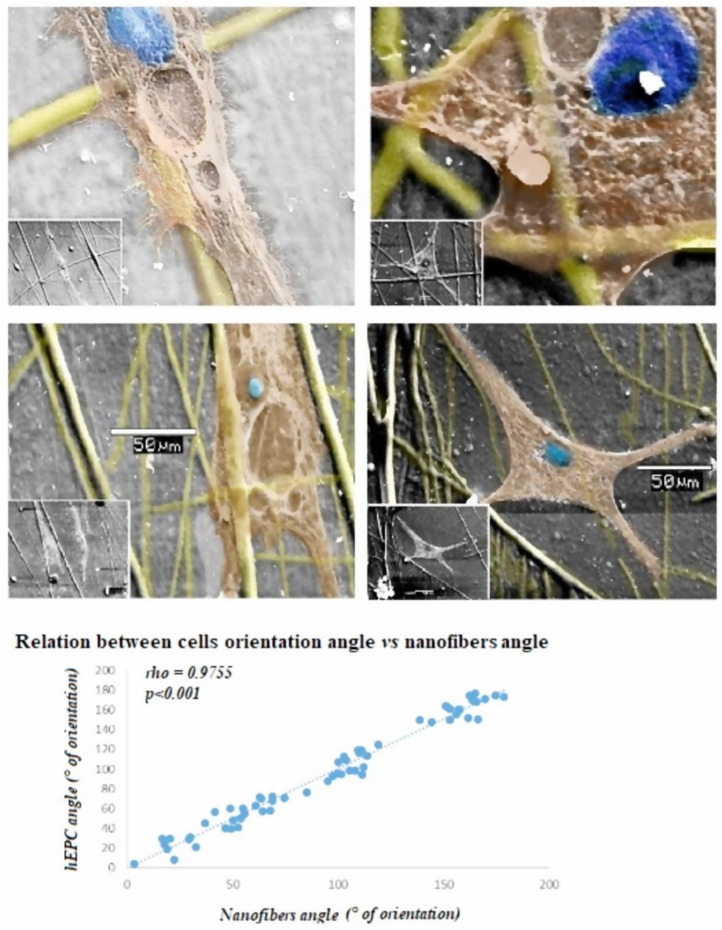
**SEM images of EPCs integrated into scaffolds.** Microphotograph showing the integration of EPCs to partially degraded collagen nanofibers (280 ± 193 nm diameter), obtained by electrospinning technique at 15 kV, 0.06 mL/h, 2000 rpm and 10 cm needle-to-collector distance working parameters. Detailed fiber–hEPCs interaction, showing cell adhesion, elongation and alignment along the fiber, are provided. Lower power field is also provided in the left corner. hEP cell integration was evaluated after 6 days of culture on DPC nanofibers. Scale bar represents 50 µm. At the bottom, an analysis of cell orientation according to scaffold is shown.

**Table 1 polymers-14-02448-t001:** Electrospinning parameters and fiber characteristics.

	Voltage (kV)	Collector to Needle Distance (cm)	Spin Speed (rpm)	Infusion Rate (mL/h)	Fiber Diameter (nm)	Mean Difference *p*-Value
Voltage and infusion rate variation	8	10	1000	0.8	No fiber obtained	NS
				0.6	No fiber obtained	
				0.06	No fiber obtained	
	10			0.8	743 ± 355	<0.05
				0.6	692 ± 152	
				0.06	703 ± 131	
	15			0.8	443 ± 295	<0.05
				0.6	307 ± 130	
				0.06	267 ± 122	
Voltage and distance variation	10	10	1000	0.06	805 ± 468	<0.05
		12			604 ± 326	
	15	10			403 ± 286	<0.05
		12			333 ± 130	
Voltage and rotation speed variation	10	10	1000	0.06	815 ± 510	<0.05
			2000		281 ± 193	
			3000		206 ± 136	
	15		1000		320 ±154	<0.05
			2000		280 ± 193	
			3000		200 ± 115	

Mean difference was determined by ANOVA test, post hoc analyse. NS, non-significant.

## Data Availability

The datasets generated and analyzed during the current study are not publicly available due to privacy policies of the hospital, but are available from the corresponding author on reasonable request.

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
