# Peer review of "Electrospinning-Generated Nanofiber Scaffolds Suitable for Integration of Primary Human Circulating Endothelial Progenitor Cells"

_polymers, 2022, doi:10.3390/polym14122448_

Round 1

Reviewer 1 Report

Accept the manuscript in this form 

Author Response

We appreciate the decision of reviewer 1

Reviewer 2 Report

In this work Jiméez-Beltrán and colleagues optimized the production of collagen electrospun fibers for the cultivation of patient-derived endothelial cells.  Overall, the content of the manuscript and the experimental results are interesting.  However, the novelty of this work is limited.  Since the last version, the authors have greatly improved the quality of the manuscript, but this still requires some refinement before publication.  I advise the manuscript to be returned to the authors with Major revisions.  Please let the authors know to consider the following in order to improve their work:

  1. In table 1, condition 15kV, 12cm, 1000rpm, 0.06mL h-1 has yielded fibers with larger diameters than those obtained from 15kV, 10cm, 1000rpm, 0.06mL h-1. Please, do the following:

1.1. Confirm these and all sets of values inserted in table 1;

1.2. Evaluate statistically significant differences between fiber diameter using ANOVA + post-hoc test.

  1. The authors were not able to perform a physico-chemical characterization of these fibers. As such, they have decided to classify these as “partially-degraded collagen fibers”. Please, consider the following:

2.1. Material characterization techniques are important for initial material/ADME development stages, such as in the work presented in this manuscript.  Please provide data to support your claim about the materials you are using.

2.2. Please change the tittle to suit the already performed changes.

  1. The authors did not present a TCP control for the cell proliferation curve. Moreover, it is not clear from which sample such curve was obtained from. Please address this, and ideally provide growth curves for the fibers and controls used.
  2. For figure 5, add in the materials and methods section the software and protocol used for image manipulation. Moreover, add the original images in SI.
  3. Please, elaborate more on the discussion section: What are the potential applications of such fibers, namely on vascular tissue engineering?

Author Response

Thank you for the comments

The response to each comment is provided in the attached file.

Reviewer 3 Report

The authors took in consideration my comments and the manuscript can now be accepted for publication. 

Author Response

We appreciate the decision of reviewer 3

This manuscript is a resubmission of an earlier submission. The following is a list of the peer review reports and author responses from that submission.

Round 1

Reviewer 1 Report

According to the manuscript with title: “ELECTROSPINNING-GENERATED COLLAGEN NANO- 2 FIBERS SCAFFOLDS SUITABLE FOR INTEGRATION OF 3 PRIMARY HUMAN CIRCULATING ENDOTHELIAL PRO- 4 GENITOR CELLS". The submitted work is introducing a new valuable and interesting idea and the given results confirm the idea. This work is suitable for publication in the Journal. I suggest the acceptance after some corrections as follows;

  1. Figure 3 is low resolution
  2. Give some results with numbers in conclusion
  3. Don’t need to add figure 1, all researchers know the Description of the electrospinning system
  4. The caption of figure 4 need to be rewrite in suitable form
  5. The paper need XRD and EDX, also, need FT-IR analysis
  6. Add more explanation to experimental work
  7. Reformulate the novelty of the work in introduction
  8. The bibliography needs to be improved. Some papers in literature could be taken into consideration such as; Wound dressing properties of functionalized environmentally biopolymer loaded with selenium nanoparticles; Differentiation between cellulose acetate and polyvinyl alcohol nanofibrous scaffolds containing magnetite nanoparticles/graphene oxide via pulsed laser ablation technique for tissue engineering applications; --It is good to mention and add all these articles that could be important in the introduction section and add to references section
  9. Correct typographical errors.
  10. Don’t use abbreviations in title and abstract, you must define it in first time use

Reviewer 2 Report

In this work Jiméez-Beltrán and colleagues optimized the production of collagen electrospun fibers for the cultivation of patient-derived endothelial cells.  Overall, the content of the manuscript and the experimental results are interesting.  However, the novelty of this work and the experimental design used are outdated and require much more work before the manuscript can be even considered for publication.  I advise the manuscript to be rejected and returned to the authors.  Please let the authors know to consider the following in order to improve their work:

OVERALL MANUSCRIPT

1 – Please revise this manuscript with a native speaker of English.

2 – Also consider revising the text to improve clarity and objectivity.

INTRODUCTION

3 – The supporting literature is outdated (most recent article in the reference list dates from 2015).  Please, update the introduction with the most recent literature on 1) electrospinning, 2) Collagen, and 3) tissue engineering.

RESULTS

4 – Image numbering should follow the order by which they are referenced in the text.  Please correct this.

5 – Regarding the fiber diameter results, systematize the fiber diameter values (average ± std) in a table for better understanding of the data.

6 – Please provide the n-value for the number of fibers used to evaluate their diameter distribution.

6 – Figure 4A appears to have been acquired by a different method than the remaining ones.  Please provide a replacement for Figure 4A.

7 – Collagen is a hard material to work with, as it is prone to denaturate into gelatin.  This was demonstrated in the work of Zeugolis and colleagues (Electro-spinning of pure collagen nano-fibres – Just an expensive way to make gelatin?, Biomaterials, 2008), where the authors have also showed that the solvent has a critical role in the process.  Please provide evidence that the fibers produced in this work are indeed composed of collagen.  For this, I propose the authors to present, at least, the data: a) TEM images of the fibers, where the triple-helix is evidenced, b) FTIR spectra of the fibers versus collagen and gelatin control samples, and c) DSC thermograms of the fibers versus collagen and gelatin control samples.  The authors are also free to choose other methods to perform such characterization.  Some other examples are present in the work of Zeugolis and colleagues (Electro-spinning of pure collagen nano-fibers – Just an expensive way to make gelatin?, Biomaterials, 2008), Chen and colleagues (Electrospun collagen–chitosan nanofiber: A biomimetic extracellular matrix for endothelial cell and smooth muscle cell, Acta Biomaterialia, 2010), and Kang and colleagues (Hyaluronic acid oligosaccharide-modified collagen nanofibers as vascular tissue-engineered scaffold for promoting endothelial cell proliferation, Carbohydrate Polymers, 2019).  Please, also check the review paper written by Blackstone and colleagues (Collagen-Based Electrospun Materials for Tissue Engineering: A Systematic Review, Bioengineering, 2021) for further references.

8 – Please replace the immunofluorescence images in Figure 2A with ones with more contrast.

9 – Please change the graphic presented in Figure 2B: replace the caption in Spanish for English.

10 – Regarding Figure 2B, please provide a control condition in TCP for comparison.

11 – Regarding Figure 2B, please insert in the materials and methods section the explanation and respective formulas for the calculation of a) viability index, b) specific growth speed and c) doubling time.

12 – Regarding Figure 5, the provided SEM images evidence sample charging.  Please provide new and more clear images.

13 – Regarding Figure 5, can the authors provide images for the contact points between EPCs and the fibers?

DISCUSSION

14 – Please expand the discussion section.  What are the main contributions of this work for the expansion of knowledge and the tissue engineering field?

Reviewer 3 Report

  • The work presented in the manuscript is of interest, but the authors fail in presenting the main novelty of the work. Is the novelty the ES of collagen or the use of collagen scaffolds to integrate EPCs?  How this work differs from the others in literature? The introduction should focus the main advances of the work regarding the current state of the art. 
  • The authors should explain in more detail the process for the integration of the cells in the collagen scaffold.  The data presented in Figure 5 corresponds to which time of contact?
  • It would be of interest to have the mechanical properties of the scaffolds, along with other properties like in vitro degradation and swelling capacity.